# Dual Glyoxalase-1 and β-Klotho Gene-Activated Scaffold Reduces Methylglyoxal and Reprograms Diabetic Adipose-Derived Stem Cells: Prospects in Improved Wound Healing

**DOI:** 10.3390/pharmaceutics16020265

**Published:** 2024-02-13

**Authors:** Nadia Pang, Ashang L. Laiva, Noof Z. Sulaiman, Priya Das, Fergal J. O’Brien, Michael B. Keogh

**Affiliations:** 1Tissue Engineering Research Group—Bahrain, Royal College of Surgeons in Ireland, Adliya P.O. Box 15503, Bahrain; nadiapang@gmail.com (N.P.); nsulaiman@rcsi.com (N.Z.S.); pdas@rcsi.com (P.D.); 2Tissue Engineering Research Group, Department of Anatomy and Regenerative Medicine, Royal College of Surgeons in Ireland, 123 St. Stephen’s Green, D02 YN77 Dublin, Ireland; 3Advanced Materials and Bioengineering Research Centre, Royal College of Surgeons in Ireland and Trinity College Dublin, D02 PN40 Dublin, Ireland

**Keywords:** glyoxalase-1, β-klotho, gene-activated scaffold, methylglyoxal, stem cells rejuvenation, anti-fibrotic, matrix deposition, adipose-derived stem cells, wound healing

## Abstract

Tissue engineering approaches aim to provide biocompatible scaffold supports that allow healing to progress often in healthy tissue. In diabetic foot ulcers (DFUs), hyperglycemia impedes ulcer regeneration, due to complications involving accumulations of cellular methylglyoxal (MG), a key component of oxidated stress and premature cellular aging which further limits repair. In this study, we aim to reduce MG using a collagen-chondroitin sulfate gene-activated scaffold (GAS) containing the glyoxalase-1 gene (GLO-1) to scavenge MG and anti-fibrotic β-klotho to restore stem cell activity in diabetic adipose-derived stem cells (dADSCs). dADSCs were cultured on dual GAS constructs for 21 days in high-glucose media in vitro. Our results show that dADSCs cultured on dual GAS significantly reduced MG accumulation (−84%; *p* < 0.05) compared to the gene-free controls. Similar reductions in profibrotic proteins α-smooth muscle actin (−65%) and fibronectin (−76%; *p* < 0.05) were identified in dual GAS groups. Similar findings were observed in the expression of pro-scarring structural proteins collagen I (−62%), collagen IV (−70%) and collagen VII (−86%). A non-significant decrease in the expression of basement membrane protein E-cadherin (−59%) was noted; however, the dual GAS showed a significant increase in the expression of laminin (+300%). We conclude that dual GAS-containing Glo-1 and β-klotho had a synergistic MG detoxification and anti-fibrotic role in dADSC’s. This may be beneficial to provide better wound healing in DFUs by controlling the diabetic environment and rejuvenating the diabetic stem cells towards improved wound healing.

## 1. Introduction

Diabetic foot ulcers (DFUs) are a progressive pathological condition [1,2] and major socioeconomic burden caused by non-healing chronic wounds of diabetes mellitus. Patients with diabetes mellitus have a 25% lifetime risk for developing DFU [3]. It has been reported that almost a quarter of DFU patients require lower limb amputation due to severe gangrene [4], which significantly elevates the risk of mortality in diabetic patients. Poor glycemic control or a hyperglycemic microenvironment is one of the most significant risk factors that accelerate ulcer recurrence, which affects 40% of DFU patients in the first year after healing [5].

The treatment approach for DFU currently includes surgical debridement, dressings aiding a moist wound environment, vascular assessment, glycemic control and treating the active infection. Various methods have been assessed to improve these approaches for DFU wound healing using natural substances (manuka honey), commercial products (jelonet^®^), platelet-rich plasma (PRP) and cellular-based therapies using mesenchymal stem cells [1,6,7,8].

Adipose-derived stem cells (ADSCs) represent mesenchymal stem cells derived from the mesoderm and ADSC tissue regeneration therapy has shown promising results in comparison to other stem cells due to low donor site morbidity rates, fewer incidences of complications, their potential to differentiate into different cell types and wound healing cytokine production [9,10,11,12]. ADSCs have been widely used in tissue engineering and have shown promising results both in vitro and in vivo in reducing fibrosis, wherein stem cell-conditioned media have been reported to inhibit the TGF-β1-induced differentiation of keloid and hypertrophic scar-derived fibroblasts [13,14]. However, in a hyperglycemic environment, dADSCs shows greater senescence with poor angiogenic properties resulting in slower wound activation in DFUs [1,15]. Diabetic stem cells display impaired cellular homeostasis wound healing with increased inflammation and fibrosis as well as reduced remodeling [15,16]. This may be attributed to the diabetic niche environment. For example, a hyperglycemic environment leads to the accumulation of cellular methylglyoxal (MG). MG is a highly reactive α-dicarbonyl compound generated during the glycolytic pathways. It has an exponentially high glycation capacity and mainly reacts with lipids, arginine residues of proteins and nucleic acids to form advanced glycation end products (AGEs) [17,18]. The elevated level of methylglyoxal has been linked to oxidative stress by the formation of reactive oxygen species (ROS) in a variety of cultured cell types [12,19] as well as diabetic retinopathy, neuropathy and cardiovascular disease [20,21,22,23]. Additionally, AGEs have been related to the increased stiffness of tissues seen in diabetes mellitus because they are able to cross-link glycated collagen irreversibly and make it resistant to enzymatic proteolysis and subsequent degradation [18]. Other reports indicate that laminin and fibronectin are more susceptible to modifications by methylglyoxal, leading to failure in sensory nerve regeneration [17]. These reports suggest that the hyperglycemic environment promotes a more fibrotic environment in the presence of methylglyoxal and AGEs; therefore, reducing MG may be beneficial to wound healing.

The detoxifying enzyme Glyoxalase-1 (GLO-1) is of particular interest to control MG as it is one of the glyoxalase systems that can detoxify excess MG and reduce AGEs’ formation [23,24,25,26]. However, in diabetes, GLO-1 expression has shown to be reduced. The reduced capacity of GLO-1 to detoxify MG has been observed in the case of the endothelial dysfunction that precedes the pathogenesis of diabetes-associated micro- and macrovascular complications [19,23,27].

Other than controlling the local MG level, it is important to direct the local wound site cells towards an anti-fibrotic, wound-healing state. For this, many natural biomaterials including alginate, collagen, chitosan, dextran, fibrin, hyaluronic acid [8] and pectin have been used due to their excellent biocompatibility. Previously in our group, we optimized a collagen scaffold for skin wound healing [1,28,29]. Briefly, bovine collagen type 1 and chondroitin sulfate from shark cartilage is blended and freeze-dried into a sponge scaffold which has shown to support many cell types. We have also tailored our collagen scaffold with anti-fibrotic pro-rejuvenating wound healing genes like β-klotho to produce a gene-activated scaffold. We have shown the possibility of reprogramming dADSCs transiently to healthy-like cells by priming β-klotho on pro-angiogenic collagen-gene activated scaffolds [19,23,30], which revealed a great potential of stem cell therapy for treating unhealed dermal wounds in diabetic patients. Klotho overexpression or supplementation have shown protection against fibrosis in various models of renal and cardiac fibrotic disease [27,31]. The Klotho family is comprised of α-klotho, β-klotho and ϒ-klotho. β-klotho is expressed as the most in adipose tissue and the pancreas. It plays an important role in promoting glycolysis and glucose-stimulated insulin secretion [32]. The anti-aging and anti-fibrotic function of soluble klotho is mainly modulated by the TGFβ1, Wnt, IGF1 and FGF2 signaling pathways which may be beneficial to reactivate stem cells in the treatment of diabetic wounds [19,23,33,34].

In this study we hypothesize that autologous dADSCs primed to reduce MG accumulations by using GLO-1 along with antifibrotic β-klotho would enhance diabetic wound healing. As an extension to our previous study, we aim to assess the combinatorial efficacy of anti-diabetic GLO-1 and anti-aging β-klotho incorporated on our porous bilayered collagen-chondroitin sulfate scaffolds. The dual GAS developed in this study was then assessed to see if lowering the pro-diabetic MG and pro-fibrotic markers of dADSC would enable it to drive towards a rejuvenating mode, which can be applied for scarless recovery in later wound healing stages.

## 2. Materials and Methods

### 2.1. Preparation of Gene-Activated Scaffold (GAS)

GAS was developed using a two-step process as defined previously [35,36,37,38]. Bovine tendon type 1 collagen and shark cartilage chondroitin-6-sulfate (Sigma, Edinburgh, UK) scaffolds were prepared through an optimized freeze-drying process [37], then treated at 105 °C under vacuum for sterilization and scaffold crosslinking [38], followed by further chemical cross-linking with 14 mM N-(3-Dimethylaminopropyl)-N′-ethylcarbodiimide hydrochloride and 5.5 mM N-Hydroxysuccinimide (EDC/NHS, Sigma, UK) solution to enhance their mechanical stability. The cross-linked scaffolds were then washed with PBS (Gibco, London, UK) before soak-loading polyplex nanoparticles on to the scaffolds for 40 min. Based on previous studies [35,36,39], polyplex particles were formulated by mixing a specified amount of branched cationic 25 kDa polyethyleneimine (PEI) (Sigma-Aldrich, Dublin, Ireland) for anionic pDNA (fixed at a dose of 2 μg) delivery [28,36,40] and given an N/P ratio of 10. Three scaffold groups are prepared: (1) gene-free scaffold (GFS, no pDNA), (2) β-klotho gene-activated scaffold and (3) Glyoxalase-1/β-Klotho gene-activated scaffold (dual GAS). The Glyoxalase-1 (Cat no. sc-401914-ACT) and β-klotho plasmids (Cat no. HG10568-UT) were obtained from SantaCruz Biotechnologies, Santa Cruz, CA, USA, and SinoBiological, Beijing, China, respectively.

### 2.2. Cell Seeding

Type 2 diabetic adipose-Derived Stem Cells (dADSCs, Lonza Bioscience, Walkersville, MD, USA, 74 YO/F/Cat no. PT-5008) were expanded to passage 4 in Dulbecco’s Modified Eagle’s Medium—high glucose with 4500 mg/L (Cat no. D5796, Sigma, UK). Then, 5 × 10^5^ dADSCs (2.5 × 10^5^ per side) were seeded for each scaffold on 12-well plates for 20 min. Next, 2 mL of transfection media OptiMEM (Gibco, UK) was added per well and incubated at 37 °C for 24 h. Cell-seeded GAS were transferred into new plates and fed with 2 mL of fresh high-glucose medium. Media changes were performed every 3–4 days until day 21.

### 2.3. Immunofluorescence Imaging

The cellular and ECM protein expressions on GAS were determined by immunofluorescence imaging. As described previously, the scaffolds were first washed with PBS and fixed in 10% neutral buffered formalin for 20 min, then processed according to the standard protocol for paraffinization. The blocks were cut into 7 μm thicknesses and collected on charged slides. After that, the sections were deparaffinized and then permeabilized with 0.2% Tween^®^20 (Sigma-Aldrich, Saint-Quentin-Fallavier, France) solution in PBS for 30 min (10 min wash × 3) and blocked using 10% NGS (Normal Goat Serum, Invitrogen, Milton, UK)/5% BSA/0.3 M Glycine (prepared in permeabilizing solution) for 1 h. After blocking, the slides were rinsed with PBS and incubated at 4 °C overnight with the antibodies to target matrix proteins listed in Table 1.

After primary antibody incubation, the slides were rinsed by PBS and then covered in either Alexa 488-conjugated goat anti-mouse IgG (Cat no. A32723, Invitrogen, UK) and/or Alexa 594-conjugated goat anti-rabbit IgG (Cat no. A11012, Invitrogen, UK) at 1:800 dilutions at room temperature for 1 h in the dark. The rinsing steps were performed as previously described and counterstained for nuclei using the mounting medium with DAPI (ab104139, Abcam, UK). The images were taken under a fluorescence microscope (Olympus BX43, Japan) at 200× magnification. Samples were incubated with only secondary antibodies as controls.

### 2.4. Image Quantification

Images were captured by cellSens Imaging Software (Cellsens Standard v1.16, Evident, Olympus Life Science Solutions, Tokyo, Japan) and semi-qualified by ImageJ. A constant threshold value of each marker was pre-adjusted and determined through preliminary imaging of various sections. Integrated density (stained area × mean gray value) of the images was measured and normalized to the number of cells (DAPI counting) to determine a final mean fluorescence density per cell. An average was quantified from at least 3 random images per non-consecutive triplicate section (*n* = 3) per group. Relative expressions between the groups were calculated for analysis.

### 2.5. Statistical Analysis

All results are expressed as mean ± standard deviation. An unpaired two-tailed *t*-test was used to demonstrate the statistically significant difference between groups, where *p* < 0.05 (marked with * and ** if *p* < 0.01) was significant. All statistical tests were performed using SPSS v 26 (IBM, US) and the graphical visualizations were generated using GraphPad PRISM 10 (GraphPad Software, San Diego, CA, USA).

## 3. Results

### 3.1. Dual GAS Reduces Methylglyoxal Production of dADSCs

GFS demonstrated the highest level of methylglyoxal expression on day 21. A pronounced decrease in all GAS was noted (Figure 1), particularly dual GAS, which demonstrated significantly reduced (84%) methylglyoxal expressions (*p* < 0.05) compared to that of GFS, remarkably suppressing the major driving factor for forming hyperglycemia-associated AGE.

### 3.2. Dual GAS Reduces the Expression of Pro-Fibrotic Markers

GFS demonstrated abundant extracellular expressions of pro-fibrotic fibronectin and scar-associated α-Smooth muscle actin (α-SMA) on day 21 (Figure 2). In contrast, all GAS demonstrated a decreasing trend of fibronectin and α-SMA depositions relative to GFS. Dual GAS demonstrated significantly (*p* < 0.05) lower fibronectin and α-SMA expressions which dropped by 76% and 65%, respectively, compared to that of GFS.

### 3.3. GAS Shows Decrease in Collagen

Collagen IV’s deposition as a fibrous network occurred in all groups and predominantly on GFS (Figure 3). All GAS deposited less collagen IV on Day 21, dual GAS deposited the least collagen IV with a decrease of 70% (*p* < 0.05) compared to GFS. Suppression was also noted on the dermal structure proteins collagen I (62%, *p* < 0.05) and collagen VII (86%, *p* < 0.05).

### 3.4. GAS Shows Contradictory Results in Basement Membrane Regeneration

Having observed the GAS’ controlled stimulation of dADSCs’ functionality, we ultimately investigated dADSCs’ response towards the regeneration of basement membrane proteins. GFS minimally expressed predominantly cellular nascent basement membrane protein laminin (Figure 4). Contrarily, dual GAS demonstrated the robustly enhanced deposition of the extracellular laminin matrix, which was 300% surged compared to that of GFS. As for the main transcellular adhesion protein of the epidermis, dual GAS showed a 59% downregulation of E-cadherin deposition compared to GFS. The presence of β-Klotho inhibited E-cadherin expression on day 21.

## 4. Discussion

Wound healing is one of the most complex processes in the human body. It involves the spatial and temporal synchronization of a variety of cell types, extracellular matrix proteins (ECM) and growth factors with distinct roles in the phases of hemostasis [44]. Stem cells serve as vehicles for the delivery of growth factors and cytokines to the wound site, thereby promoting the migration of other cell types which transiently stream through phases of wound healing [13,14]. A hyperglycemic environment impairs wound healing as well as mediates the diabetic stem cells to a senescent stage with dADSCs displaying poor angiogenic properties [12,16]. Similarly, a hyperglycemic environment inducing cellular methylglyoxal (MG) accumulation can impair wound healing by a combination of localized stem cell desensitization and cause protein aggregation, fibril formation and protease resistance [45]. Therefore, the aim of this study was to assess if a dual GAS containing GLO-1 and β-klotho could switch the senescent dADSC’s channeling to improve diabetic-based wound healing.

The study results show a statistically significantly reduction in the methylglyoxal of dADSCs in hyperglycemic conditions with the dual GAS (reduction of 84%). Our results are in line with Peng et al. who showed that the enforced expression of the GLO-1 gene in dADSCs using lentiviruses restored the rejuvenate properties of the dADSCs [30]. GLO-1 overexpression restored the proangiogenic capacity and has been reported to prevent vascular aging and reverse hyperglycemia-induced angiogenic defects in human endothelial cells, bone-marrow-derived stem cells and cardiac stem cells [23,27,46]. In support for the role of protein glycation, Molgat et al. observed that somatic gene transfers of GLO-1 restored the angiogenic capacity of diabetic cardiac stem cells. Interestingly, GLO-1 overexpression also returned the production of the proinflammatory cytokine interleukin-6 to nondiabetic levels, which may suggest a critical role in cardiac stem cell-mediated repair [47].

Scavenging MG is an effective method in reducing MG-mediated stress and macrophage dysfunction, thereby promoting granulation and wound healing [48]. The overexpression of GLO-1 in endothelial cell lines has increased the protection against a hyperglycemia-induced angiogenesis deficit by reducing MG [49]. We also show that the therapeutic dual GAS containing GLO-1 and β-klotho genes regenerated dADSCs through remodeling them by reducing MG and enhancing anti-fibrotic and decreasing fibrotic basement proteins.

One of the interesting findings of this study which has not been reported elsewhere in the literature is the decrease of MG in dADSCs treated with only β-klotho gene-activated scaffolds. The overexpression of klotho has shown to increase longevity by inhibiting the IGF-1/insulin pathway [33]. However, the exact mechanism by which β-klotho reduces MG requires further investigation.

In our previous study, we showed that β-klotho transiently enhanced ADSC’s stemness and the early activation of anti-fibrotic genes TGF- β3, which are essential for controlled healing with reduced scarring [35]. This effect was also shown to improve dADSC’s with β-klotho GAS [36]. A similar response was noted in this study with klotho GAS; however, we also note a synergistic response in our dual GAS to reduce the profibrotic markers fibronectin and α-SMA.

MG stress due to GLO-1 depletion has shown ECM modulation in cells. The knockdown of GLO-1 in human aortic endothelial cells have shown an increase in the extracellular concentrations of Type I collagen [50]. Reports indicate that GLO-1 modulates TGF- β/Smad signaling to increase the levels of matrix metalloproteinase 2 and 9, endopeptidases required for ECM degradation [51]. Collagen glycation by MG plays a critical role in the fibrosis and high expression of the pro-fibrotic markers α-SMA, and fibronectin and cadherin were reported in cells grown on methylglyoxal-treated collagen [52]. The controlled expression of α-SMA and fibronectin is required in the repair of injured tissue as prolonged activity may result in fibrosis and organ dysfunction [53]. Fibronectin is a glycoprotein, and its assembly is an important initial step for the deposition of other ECM proteins, including collagen. It acts with other proteins in the wound bed to allow cell matrix assembly and the deposition of granulation tissue followed by its degradation. However, in chronic wounds like DFU, fibronectin degradation is imbalanced which mediates a fibrotic niche [54]. Myofibroblasts or activated fibrogenic cells increase the expression of α-SMA which causes scar contraction. Klotho-derived peptides have been shown to block fibronectin, collagen-1 and α-SMA expression induced by the TGF- β (Smad-dependent and -independent) signaling pathway in various type of cells [55]. The results of this study agree with these findings where fibronectin and α-SMA were reduced in dADSCs primed with klotho-GAS only and with dual GAS as well.

There is continuous collagen synthesis and collagen degradation in the normal condition, and these two processes are precisely balanced to maintain normal tissue architecture and are critical in preventing the formation of permanent scar tissue. Collagen fibers promote wound contraction which is indispensable for complete wound closure [56,57]. During granulation tissue maturation, the excess collagen degrades, and wound contraction also begins to peak around Day 21. However, an aggressive sustained increase in collagen I deposition can increase scar formation [57]. Type I collagen is a fibrillar protein and is the most abundant type in the tissues. Collagen IV is mainly found in the dermal–epidermal junction. It is structurally more pliable and forms sheets in the cutaneous basal lamina. Collagen I and IV act as chemoattractant for neutrophils and hence promote the inflammation phase in normal wound healing. Collagen VII, in the dermal–epidermal junction zone serves, as an adhesion molecule mainly for the migration of keratinocytes. Collagen VII has shown to affect the expression of integrins binding to laminin during the re-epithelization of the wound [58].

Controlling scarring is crucial as it can inhibit the development of other skin appendages such as hair follicles, sebaceous glands and sweat glands, for example, in burn wounds [59]. ADSC has proved to reduce collagen I deposition in various organs [60]. In our previous study, we showed that fibroblasts stimulated with day 14 conditioned media from ADSCs on GAS showed a reduced expression of pro-fibrotic collagen I [36]. Li et al. have also shown that the ADSCs-conditioned media can significantly reduce collagen I expression in fibroblasts derived from hypertrophic scars [61]. Another study by Wang et al. demonstrated a similar reduction in collagen I expression in keloid-derived fibroblasts [62], collectively demonstrating the anti-fibrotic potential of ADSCs’ conditioned media.

Our study aligned with these studies and indicated that the β-klotho GAS and dual GAS transiently regenerates the function of dADSC in reducing pro-fibrotic collagen-1 by reducing the MG. Our findings demonstrate that dual GAS offered better control in limiting scarring in vitro through controlled collagen I expression.

The lack of a correlation of basement membrane protein collagen IV and the reduction of dermal–epidermal basement membrane protein collagen VII could be attributed to the rate of wound remodeling [58,59]. The greatest rate of the accumulation of connective tissue within the healthy wound occurs 7–14 days after injury, which translates into the period of the most rapid gain in tensile strength. Thereafter, the collagen content within the wound levels off as fibroblasts down-regulate their synthetic activities; this corresponds to a much slower gain in tensile strength as the wound remodels [56]. That could also be explained by the fact that the maturational or remodeling phase of normal wound healing starts around Day 21. However, up to now many studies investigate the biological mechanism of collagen I and collagen III, but the involvements of collagen IV and VII in later wound healing stage are not clearly understood. Further research should further develop and confirm the biological mechanisms of different collagens’ involvement in the remodeling phase.

Finally, we investigated the dADSCs response towards the regeneration of basement proteins with dual GAS. Laminins are important structural components of the basement membrane that contribute to cell migration, attachment and wound healing. It is one of the first basement membrane components laid down during injury. Laminin promotes keratinocyte migration to the site of the wound and re-establishes the epithelial barrier and anchorage by serving as an adhesive ligand in the basement membrane. The increased expression of laminin is seen as the earliest event of re-epithelization, with expression in epidermal keratinocytes on the leading edge of the wound [63]. Recent wound healing studies have indicated potential migratory and angiogenic properties elicited by peptides present on the α and γ chains of laminin. Changes in laminin expression have been identified during normal wound repair and defects in this distribution have been correlated with delayed or impaired wound closure [64]. Laminin is highly susceptible to glycation by MG and these AGE products have contributed to the failure of nerve regeneration in diabetic neuropathy [65]. The results of this study are in line in our previous study [36] where the sole priming of dADSCs with β-klotho showed an increased production of laminin. The increase in the deposition of the basement membrane protein laminin on β-klotho GAS and dual GAS on day 21 also reflected the juvenile effect of β-klotho which enhances basement membrane repairing over a longer period. The finding supported that dual GAS facilitates laminin deposition for a more normalized wound repair. This can be attributed to GLO-1 increasing the metabolism of MG and providing a microenvironment more in favor of basement membrane repairment.

Moreover, both β-klotho GAS and dual GAS showed the downregulation of E-cadherin depositions. E-cadherin is a potent cell–cell adhesion molecule in epithelial tissue which plays a role in cellular adhesion and motility. During wound healing, the cells at the top and basal layer of the epithelia showed decreased E-Cadherin expression during migration and mitosis or endocytosis [66]. Our results were supportive for a controlled wound healing environment; on the contrary, Ibi et al. reported that the overexpression of klotho did not affect the regulation of E-cadherin [67]. Therefore, further studies are needed to determine how E-cadherin expression is regulated and if its downregulation by β-klotho and GLO-1 genes would be beneficial in controlled wound healing.

FDA-approved Integra ™ Life Sciences collagen–glycosaminoglycan biodegradable scaffolds are widely used for the restoration of soft tissue loss [68]. It has been reported that the wound healing ability of these biocompatible materials can be enhanced by incorporating bioactive molecules (DNA/RNA/peptides/inorganic molecules) and thereby providing intricate control on the specific pathways or stages of wound healing, including diabetic wounds [1,69]. The result of this study also provides a baseline knowledge towards developing a gene-engineered dermal graft specifically to treat DFU.

## 5. Conclusions

Methylglyoxal build-up in diabetic foot ulcers can impair cell activation and limit wound healing. This study showed that a combined dual-action gene-activated scaffold (GAS) containing genes for GLO-1 and β-klotho on a collagen scaffold statistically significantly reduce toxic methylglyoxal accumulation by 84% of diabetic adipose-derived stem cells (dADSCs). The reprogrammed dADSCs became more normalized and anti-fibrotic in nature with reductions in the expression of pro-fibrotic proteins fibronectin, SMA and pro-scarring collagen I, IV and VII. The dual GAS also showed a statistically significant 300% increase in the expression of basement membrane laminin production and a decrease in E. cadherin at 21 days’ culture.

In conclusion, this study shows that the dual GAS containing GLO-1 and β-klotho genes directed a more normalized dADSC response and may be beneficial in directing improved wound healing for diabetic foot ulcers.

## Figures and Tables

**Figure 1 pharmaceutics-16-00265-f001:**
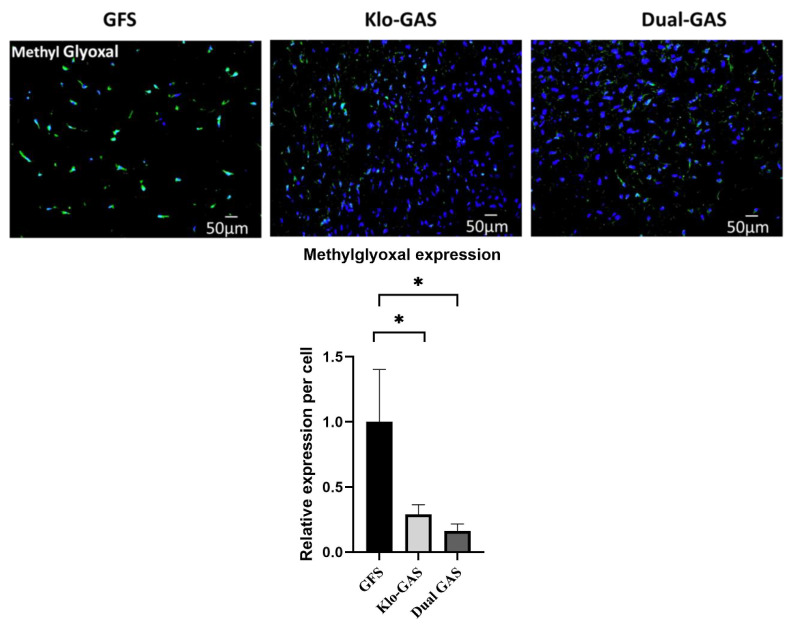
GAS demonstrated significant suppression of methylglyoxal on day 21, especially the expression of dual GAS which was notably reduced by 84% compared to GFS. Scale bar 50 µm. Data represent mean ± standard deviation (*n* = 3), * indicates *p* < 0.05.

**Figure 2 pharmaceutics-16-00265-f002:**
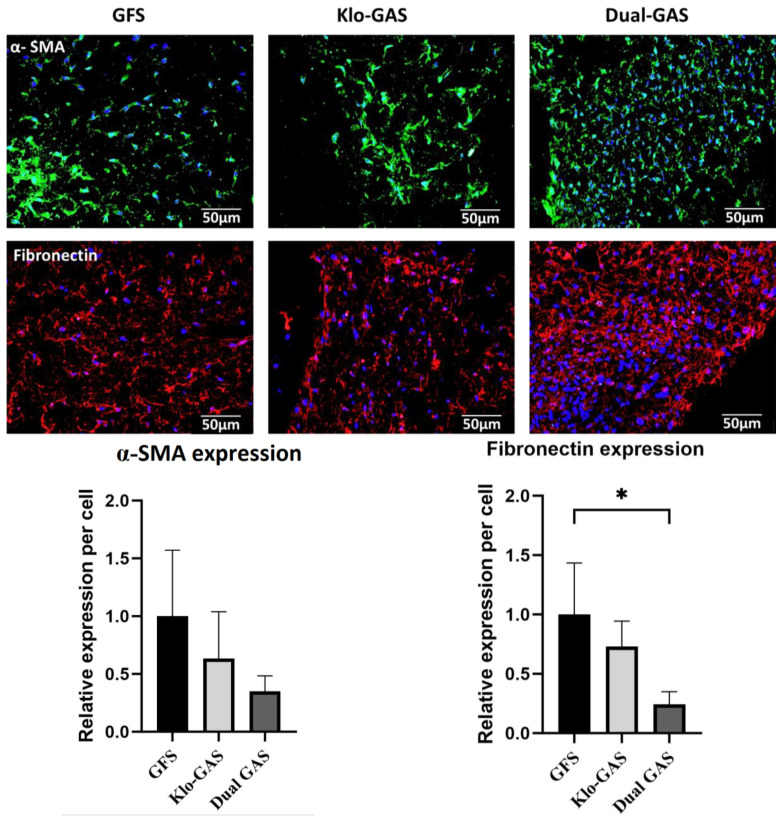
GAS expressed decreasing trend of fibronectin and α-smooth muscle actin extracellular depositions compared to GFS on day 21. The fibronectin and α-SMA expressions on dual GAS were reduced by 76% and 65%, respectively, compared to GFS. Scale bar 50 µm. Data represent mean ± standard deviation (*n* = 3) * indicates *p* < 0.05.

**Figure 3 pharmaceutics-16-00265-f003:**
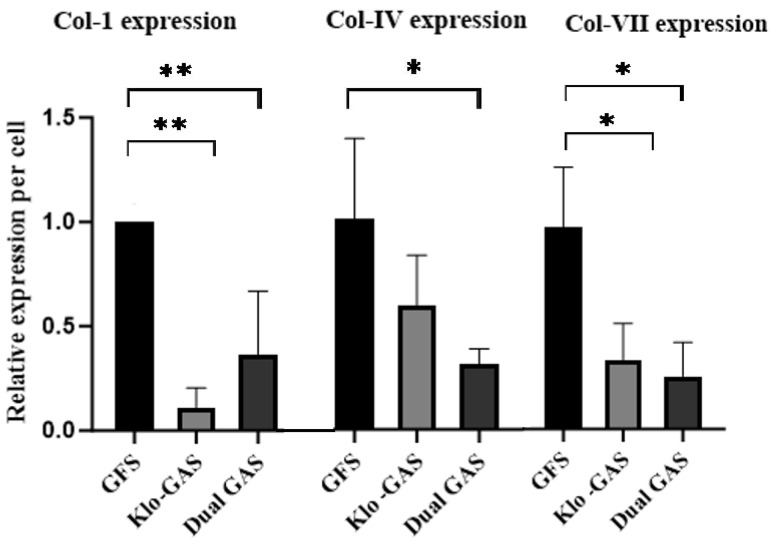
GAS showed reduced collagen expressions on day 21. Dual GAS expressed the least of Collagen IV and Collagen VII which were decreased by 70% and 86% (*p* < 0.05), respectively, compared to GFS. Also, 62% drop in Collagen I was observed on dual GAS compared to GFS, * indicates *p* < 0.05 and ** indicates *p* < 0.01.

**Figure 4 pharmaceutics-16-00265-f004:**
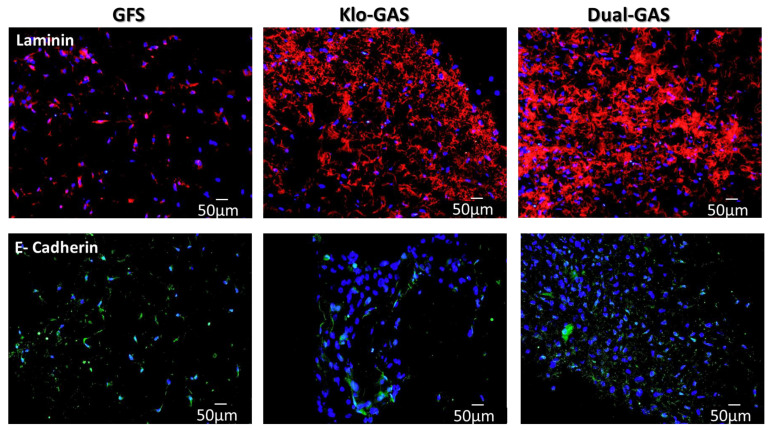
GFS expressed minimal cellular laminin, whereas dual GAS robustly enhanced 300% deposition of extracellular laminin matrix compared to that of GFS. Contrary, dual GAS showed 59% downregulations of cell–cell adhesion dimer E-cadherin deposition. Scale bar 50 µm. Data represent mean ± standard deviation (*n* = 3). * indicates *p* < 0.05.

**Table 1 pharmaceutics-16-00265-t001:** List of antibodies for assessing extracellular matrix (ECM) regeneration on GAS in reparative maturation.

Indicators	Primary Antibodies (Catalog No.)	Functional Roles	Dilutions in 1% BSA Solution
Glyoxylate	Methylglyoxal (NBP2-59368, Novus Biologicals, Abingdon, UK)	Non-enzymatic glycation of proteins; promotes diabetic vascular dysfunctions	1:100
Pro-fibrotic	Alpha-smooth muscle actin (ab7817, Abcam, Cambridge, UK)	Structural filament protein; promotes contraction and scarring	1:200
	Fibronectin (ab2413, Abcam, UK)	Provisional matrix protein; promotes fibrosis [41,42]	1:200
Basement membrane	E-cadherin (ab1416, Abcam, UK)	Mediate cell–cell adhesion; regulating contact formation and stability	1:200
	Laminin (ab11575, Abcam, UK)	Nascent basement membrane (BM) protein; BM assembly	1:200
	Collagen IV (ab6586, Abcam, UK)	Mature BM protein; BM stability [43]	1:200
Dermal matrix	Collagen I (NB600-408, Novusbio, Centennial, CO, USA)	Major body collagen; provide structure to tissue and skin	1:200
	Collagen VII (ab6312, Abcam, UK)	Epidermal basement membrane; dermal–epidermal adhesion	1:200

## Data Availability

The data presented in this study are available on request from the corresponding author. The data are not publicly available due to privacy or ethical restrictions.

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
