# Peer review of "Dual Glyoxalase-1 and β-Klotho Gene-Activated Scaffold Reduces Methylglyoxal and Reprograms Diabetic Adipose-Derived Stem Cells: Prospects in Improved Wound Healing"

_pharmaceutics, 2024, doi:10.3390/pharmaceutics16020265_

Round 1

Reviewer 1 Report

Comments and Suggestions for Authors

This study is interesting with clinical significance. DFU significantly elevates the risk of mortality in diabetic patients. The authors put forward a new and comprehensive point of view on reprograming diabetic adipose derived stem cells to improve wound healing. The followings are some comments to the authors.

Comments:

1. How to identify Type 2 diabetic adipose-Derived Stem Cells? What's the difference between normal ADSCs and dADSCs? I suggest the authors add dADSCs identification tests including primary cells and passage 4 cells.

2. What does* and  **” mean in figures? Please state it in the figure legend.

3. How can Dual glyoxalase-1 & β-klotho gene-activated scaffold be used in clinical treatment?What route of administration might be chosen?Please state it in the section of Discussion.

4. Adipose-derived stem cell is not the only cell type involved in wound healing, but other cell types as well, such as fibroblast, endothelial cell and so on. Why was dADSC chosen for this study? Please state it and explain the effect of other cell types on wound healing while using GAS treatment in the section of Discussion.

Author Response

Reviewer 1: This study is interesting with clinical significance. DFU significantly elevates the risk of mortality in diabetic patients. The authors put forward a new and comprehensive point of view on reprograming diabetic adipose derived stem cells to improve wound healing. The followings are some comments to the authors.

Comments:

  1. How to identify Type 2 diabetic adipose-Derived Stem Cells? What's the difference between normal ADSCs and dADSCs? I suggest the authors add dADSCs identification tests including primary cells and passage 4 cells.

Thank you for suggesting this. We have included the following in the first line of section 2.2 (page no. 6: line no. 155).

     “The cell lines used for this study was dADSCs from Lonza Bioscience, USA, 74 YO/F/Cat no. PT-5008”.

The product has been tested up to passage 5 and shown to express CD13, CD29, CD44, CD73, CD90, CD105, and CD166.

  1. What does“*” and  “**” mean in figures? Please state it in the figure legend.

The p value corresponding to the difference < 0.05 is denoted by * and < 0.01 is denoted by **. We have added to the methods (page 8, line no 200) and to the figure legends as suggested (Page no. 11, Fig. 3).

  1. How can Dual glyoxalase-1 & β-klotho gene-activated scaffold be used in clinical treatment? What route of administration might be chosen? Please state it in the section of Discussion.

The clinical application most suited to this gene activated scaffold is based on the traditional tissue engineering triad. Sterile scaffolds may be soak loaded with DNA plasmids prior to implantation to the wound site. This may be in the presence of autologous stem cells like diabetic ADSC’s to support localised wound healing; this was one of reason this cell type was chosen for this study.

We have added to the discussion as per reviewer’s suggestion (Page no. 17, line no 392-399). References are added in the clean version of the manuscript.

“FDA approved Integra LifeSciencesTM collagen-glycosaminoglycan biodegradable scaffolds are widely used for the restoration of soft tissue loss. It has been reported the wound healing ability of these biocompatible materials can be enhanced by incorporating bioactive molecules (DNA/RNA/peptides/inorganic molecules) and thereby providing an intricate control on the specific pathways or stages of wound healing including  diabetic wounds. The result of this study also provides a baseline knowledge towards developing a gene-engineered dermal graft in specifically to treat DFU.”

  1. Adipose-derived stem cell is not the only cell type involved in wound healing, but other cell types as well, such as fibroblast, endothelial cell and so on. Why was dADSC chosen for this study? Please state it and explain the effect of other cell types on wound healing while using GAS treatment in the section of Discussion.

Adipose derived stem cells have been shown to have excellent wound healing capabilities for soft tissue healing. They can direct other cell types in the wound bed to initiate healing while also maintaining a low inflammatory response. (Ref 9-12 from clean version).

We have stated the rational of using the dADSC in this study in the introduction section (Page 3, line no. 66- 75). The use of gene activated scaffolds and its in vitro applications has been detailed in page no 4&5, line no. 113- 118).

In addition to address to the reviewer’s comments and to further clarify it we have added a statement under discussion in page 13: line no. 272- 275)

“Stem cells serve as vehicles for delivery of growth factors and cytokines to the wound site and thereby promoting migration of other cell types leading to transiently stream through phases of wound healing [13, 14].”

Reviewer 2 Report

Comments and Suggestions for Authors

The study delves into the intricate processes of wound healing in a hyperglycemic environment, focusing on the potential of a dual gene-activated scaffold (GAS) containing Glyoxalase-1 (GLO-1) and β-klotho to improve diabetic-based wound healing. While the research provides valuable insights into the molecular and cellular aspects of wound repair, there are several points that merit critical consideration.

The study could benefit from providing a more explicit contextualization of its findings in the broader landscape of wound healing research. Addressing how these findings align with or deviate from existing literature would enhance the study's impact.

While the study touches upon the effects of GLO-1 and β-klotho on collagen and basement membrane proteins, a more in-depth mechanistic understanding of how these genes influence specific signaling pathways or cellular processes would strengthen the research.

The clinical relevance of the study could be emphasized further. Discussing how these findings might translate into potential therapeutic interventions or clinical applications for diabetic foot ulcers would enhance the practical implications of the research.

In conclusion, while the study makes notable contributions to the understanding of wound healing in a diabetic context, there is room for further contextualization, mechanistic elucidation, and exploration of clinical implications to enhance the overall impact of the research.

Author Response

Reviewer 2: The study delves into the intricate processes of wound healing in a hyperglycemic environment, focusing on the potential of a dual gene-activated scaffold (GAS) containing Glyoxalase-1 (GLO-1) and β-klotho to improve diabetic-based wound healing. While the research provides valuable insights into the molecular and cellular aspects of wound repair, there are several points that merit critical consideration. In conclusion, while the study makes notable contributions to the understanding of wound healing in a diabetic context, there is room for further contextualization, mechanistic elucidation, and exploration of clinical implications to enhance the overall impact of the research.

  1. The study could benefit from providing a more explicit contextualization of its findings in the broader landscape of wound healing research. Addressing how these findings align with or deviate from existing literature would enhance the study's impact.

As suggested by the reviewer we have included the following to provide a more contextualised description of wound healing in the introduction and discussion:

Page no. 3 (line 60- 84) :

“Treatment approach for DFU currently includes surgical debridement, dressings aiding a moist wound environment, vascular assessment, glycaemic control and treating the active infection.  Various methods have been assessed to improve these approaches for DFU wound healing using natural substances (manuka honey), commercial products (Jelonettm platelet-rich plasma (PRP) and cellular based therapies using mesenchymal stem cells [1,6-8].

Adipose-derived stem cells (ADSCs) represent mesenchymal stem cells derived from the mesoderm and ADSC tissue regeneration therapy have shown promising results in comparison to other stem cells due to low donor site morbidity rates, lesser incidences of complications, their potential to differentiate into different cell types and wound healing cytokine production [9-12]. ADSC’s has been widely used in tissue engineering and has shown promising results both in vitro and in vivo in reducing fibrosis, wherein stem cell conditioned media have been reported to inhibit TGF-β1-induced differentiation of keloid and hypertrophic scar-derived fibroblasts [13,14]. However, in hyperglycemic environment dADSCs shows greater senescence with poor angiogenic properties resulting in slower wound activation in DFUs [1,15]. Diabetic stem cells display impaired cellular homeostasis wound healing with increased inflammation and fibrosis as well as reduced remodeling [15,16]. This may be attributed to the diabetic niche environment. For example, a hyperglycemic environment leads to the accumulation of cellular methylglyoxal (MG). MG is a highly reactive α-dicarbonyl compound generated during glycolytic pathways. It has exponentially high glycation capacity and mainly reacts with lipids, arginine residues of proteins, and nucleic acids to form advanced glycation end products (AGEs) [17-19].  Elevated level of methylglyoxal has been linked to oxidative stress by formation of reactive oxygen species (ROS) in a variety of cultured cell types [12,20-21] as well as diabetic retinopathy, neuropathy, and cardiovascular disease”.

Page no. 4 (line 107-111):

 “Other than controlling the local MG level, it is important to direct the local wound site cells towards anti fibrotic, wound healing state. For this many natural biomaterials including alginate, collagen, chitosan, dextran, fibrin, hyaluronic acid [8] and pectin have been used due to their excellent biocompatibility.  Previously in our group we optimized a collagen scaffold for skin wound healing.”

Page no. 13 (Line no 270-282):

“Wound healing is one of the most complex processes in the human body. It involves the spatial and temporal synchronization of a variety of cell types, extracellular matrix proteins (ECM) and growth factors with distinct roles in the phases of haemostasis [50,51]. Stem cells serve as vehicles for delivery of growth factors and cytokines to the wound site and thereby promoting migration of other cell types leading to transiently stream through phases of wound healing [13.14]. Hyperglycaemic environment impairs wound healing as well as mediates the diabetic stem cells to a senescent stage with dADSC’s displaying poor angiogenic properties [12,16].  Similarly, a hyperglycaemic environment induced cellular methylglyoxal (MG) accumulation can impair wound healing by a combination of localised stem cell desensitisation and cause protein aggregation, fibril formation and protease resistance [51,52].  Therefore, the aim of this study was to assess if a dual GAS containing GLO-1 and β-klotho could switch the senescent dADSC’s channelizing to improve diabetic based wound healing”.

As suggested by the reviewer to provide more clarity in the literature Ref 27, 31-34 is relocated to the discussion on page 13 (line 285-293) to support our findings of reduction in MG. [Ref 23, 27, 30, 46 in the clean version of the manuscript]. Ref 45, 70-71 align with our findings that MG reduction can regenerate dADSC functions thereby reducing the pro-fibrotic collagen-1. [Ref 36, 61 and 62 in the clean version of the manuscript.

  1. While the study touches upon the effects of GLO-1 and β-klotho on collagen and basement membrane proteins, a more in-depth mechanistic understanding of how these genes influence specific signaling pathways or cellular processes would strengthen the research.

Further to reviewer’s comments, we have included the following:  

“Reducing the MG accumulations in the dADSC’s [action of GLO-1 is mentioned in page 4 (line no 92)] alters the properties of ECM proteins [as mentioned in page 14, line 305- 317). The ani-aging klotho further synergizes the ECM modulation through TGFβ1, Wnt, IGF1, and FGF2 signaling pathways (page no. 5: line no. 122-125).”

We have mentioned that further investigations are required to elucidate the in-depth cellular processes (page no. 13: line no. 294, page no. 16: line no. 388)

  1. The clinical relevance of the study could be emphasized further. Discussing how these findings might translate into potential therapeutic interventions or clinical applications for diabetic foot ulcers would enhance the practical implications of the research.

Thank you for pointing this important aspect. We have added to the discussion as per reviewer’s suggestion (Page no. 17, line no 392-399). References are added in the clean version of the manuscript.

“FDA approved Integra LifeSciencesTM collagen-glycosaminoglycan biodegradable scaffolds are widely used for the restoration of soft tissue loss. It has been reported the wound healing ability of these biocompatible materials can be enhanced by incorporating bioactive molecules (DNA/RNA/peptides/inorganic molecules) and thereby providing an intricate control on the specific pathways or stages of wound healing including  diabetic wounds. The result of this study also provides a baseline knowledge towards developing a gene-engineered dermal graft in specifically to treat DFU.”

Reviewer 3 Report

Comments and Suggestions for Authors

Dear Author's

It was very interesting work for me and I enjoyed reading it. I have no doubts that the work is valuable - it has scientific and educational value. However, I see some aspects of it that, in my opinion, require additions / improvement namely:

1. The authors present the conclusions of the examination... please comment in the discussion regarding the implications of the manuscript, how it can be used in clinical practice? 2 What may be the costs of possible therapies? Can the authors provide anything on this subject? 3 Position of references - in my opinion too much ... (some of them are over 15 years old ---history...) best regards

Author Response

Reviewer 3: It was very interesting work for me and I enjoyed reading it. I have no doubts that the work is valuable - it has scientific and educational value. However, I see some aspects of it that, in my opinion, require additions / improvement namely:

  1. The authors present the conclusions of the examination... please comment in the discussion regarding the implications of the manuscript, how it can be used in clinical practice?

Thank you for pointing this important aspect. We have added to the discussion as per reviewer’s suggestion (Page no. 17, line no 392-399). References are added in the clean version of the manuscript.

“FDA approved Integra LifeSciencesTM collagen-glycosaminoglycan biodegradable scaffolds are widely used for the restoration of soft tissue loss. It has been reported the wound healing ability of these biocompatible materials can be enhanced by incorporating bioactive molecules (DNA/RNA/peptides/inorganic molecules) and thereby providing an intricate control on the specific pathways or stages of wound healing including  diabetic wounds. The result of this study also provides a baseline knowledge towards developing a gene-engineered dermal graft in specifically to treat DFU.”

  1. What may be the costs of possible therapies? Can the authors provide anything on this subject?

The cost of FDA approved Integra is approximately 15-30 USD per square centimeter. Considering the additional cost of gene activation, the approximate cost would be around 30-50 USD per square centimeter. This cost depends on production quantity, location of fabrication.  However, this cost  would have to be compared against the reduced cost of chronic debridement’s on a healthcare system if it can provide a wound healing platform to non-union chronic DFU lesions.

3 Position of references - in my opinion too much ... (some of them are over 15 years old ---history...) 

As suggested by the reviewer, we have updated the list of references to be more recent publications where applicable. Papers more  than 15 years were either removed or replaced with two new references were added to incorporate suggestions by the reviewers.

Chang DK, Louis MR, Gimenez A, Reece EM. The Basics of Integra Dermal Regeneration Template and its Expanding Clinical Applications. Semin Plast Surg. 2019;33(3):185-9.

Tu Z, Zhong Y, Hu H, Shao D, Haag R, Schirner M, et al. Design of therapeutic biomaterials to control inflammation. Nat Rev Mater. 2022;7(7):557-74.

Please note Ref No. 29 (in clean version) was retained as it refers to the initial literature describing the preparation of the scaffold by our team (O’Brien et al., 2005).  

Round 2

Reviewer 1 Report

Comments and Suggestions for Authors

I suggest that manuscript can be accepted in present form 

Reviewer 2 Report

Comments and Suggestions for Authors

The requested changes have been made, and the manuscript has been revised accordingly